# Hyperbaric Oxygen Treatment: Effects on Mitochondrial Function and Oxidative Stress

**DOI:** 10.3390/biom11121827

**Published:** 2021-12-03

**Authors:** Nofar Schottlender, Irit Gottfried, Uri Ashery

**Affiliations:** 1School of Neurobiology, Biochemistry and Biophysics, Life Sciences Faculty, Tel Aviv University, Tel Aviv 69978, Israel; schottlender@mail.tau.ac.il (N.S.); iritgo@tauex.tau.ac.il (I.G.); 2Sagol School of Neuroscience, Tel Aviv University, Tel Aviv 69978, Israel

**Keywords:** hyperbaric oxygen treatment (HBOT), mitochondrial function, reactive oxygen species (ROS), superoxide dismutase (SOD), neuroinflammation, oxidative stress, SIRT1, HIF1a, Nrf2, hyperoxic–hypoxic paradox

## Abstract

Hyperbaric oxygen treatment (HBOT)—the administration of 100% oxygen at atmospheric pressure (ATA) greater than 1 ATA—increases the proportion of dissolved oxygen in the blood five- to twenty-fold. This increase in accessible oxygen places the mitochondrion—the organelle that consumes most of the oxygen that we breathe—at the epicenter of HBOT’s effects. As the mitochondrion is also a major site for the production of reactive oxygen species (ROS), it is possible that HBOT will increase also oxidative stress. Depending on the conditions of the HBO treatment (duration, pressure, umber of treatments), short-term treatments have been shown to have deleterious effects on both mitochondrial activity and production of ROS. Long-term treatment, on the other hand, improves mitochondrial activity and leads to a decrease in ROS levels, partially due to the effects of HBOT, which increases antioxidant defense mechanisms. Many diseases and conditions are characterized by mitochondrial dysfunction and imbalance between ROS and antioxidant scavengers, suggesting potential therapeutic intervention for HBOT. In the present review, we will present current views on the effects of HBOT on mitochondrial function and oxidative stress, the interplay between them and the implications for several diseases.

## 1. Introduction

Under normal conditions, most of the oxygen that we consume is carried by hemoglobin, which is ~98% saturated at sea-level pressure. A small fraction of oxygen that is not carried by hemoglobin is dissolved in the plasma. HBOT elevates the partial pressure of oxygen in the blood and tissues [1]. Following Henry’s Law, the percentage of dissolved oxygen increases as the partial pressure of oxygen rises. At sea level, the percentage of dissolved oxygen in the blood is 0.32, and at 2.5 ATA, it is 5.62 [1]. This 20-fold increase in dissolved oxygen in the blood reaches all body tissues, providing excess oxygen to tissues that are suffering from a lack of delivered oxygen. Hence, HBOT has been used to treat many diseases [2,3,4,5,6,7,8,9] and was shown to improve cognition in several brain disorders [10,11,12,13,14]. Although oxygen is needed for energy production in the form of adenine triphosphate (ATP), it can sometimes have deleterious effects when it interacts with other molecules, exchanges electrons and is transformed into reactive oxygen species (ROS). These ROS can damage tissues by “stealing” electrons from lipids, proteins, DNA, etc., rendering them inactive or reactive themselves. ROS have several sources and different forms, e.g., superoxide (O_2_•−), hydrogen peroxide (H_2_O_2_), hydroxyl (HO), alkoxy (RO) and more. These free radicals are cleared by enzymatic and nonenzymatic antioxidants. Enzymatic antioxidants change the ROS into nonharmful molecules (such as water, or back to oxygen) and they include superoxide dismutase (SOD), catalase, heme oxygenase 1 (HO-1), thioredoxin and glutathione-dependent peroxidase (GPx) and reductase(s) [15]. Alongside these enzymes, nonenzymatic antioxidants and endogenous radical scavengers such as vitamin C, vitamin E, glutathione, melatonin, uric acid and β-carotene also reduce ROS levels [16,17] by donating an electron to neutralize the unstable reactive species. Although HBOT has been found to elevate ROS production, especially via the mitochondria—the organelle that consumes most of the oxygen in an organism, studies have shown that HBOT also elevates antioxidant levels and activity, thereby efficiently reducing ROS levels. In the following review, we discuss the mechanism governing ROS production and the interplay between HBOT, mitochondrial function, ROS and antioxidative species.

## 2. Mitochondrial Function and Oxidative Stress

### 2.1. Mitochondrial Function

Mitochondria are found in most cells in which the biochemical process of respiration utilizes oxygen to create energy. The common endosymbiosis hypothesis claims that 1.5 billion years ago, mitochondria were infused into the cytoplasm, providing aerobic cellular respiration. Mitochondria also mediate other cellular processes and homeostatic mechanisms, such as apoptosis [18], autophagy [19], cell-cycle control [20], Ca^2+^ level regulation [21] and synaptic plasticity [22]. Mitochondria consume roughly 85–90% of the oxygen that we breathe and are not only the major source of ATP production, but also the main source for ROS generation in the cell. As HBOT elevates oxygen levels in all tissues, it is likely that its main molecular target is the mitochondrion. At the cellular level, HBOT can improve mitochondrial redox, preserve mitochondrial integrity, activate transcription factors, alleviate oxidative stress and promote neuroprotection. At the systemic level, it can mitigate harmful disease effects by reducing intracranial pressure [23], inducing angiogenesis [24] and increasing the release of neurotrophins [25].

Mitochondria are double-membrane organelles that harness most of the energy required by cells to grow, function and reproduce. The production of the cell’s energy currency, ATP, occurs at the inner mitochondrial membrane. One of the key roles of this membrane is to act as a barrier for positively charged particles—protons. This separation is used by the oxidative phosphorylation complexes (Figure 1) to pump protons from the mitochondrial matrix to the intermembrane space and create an electrochemical gradient—the mitochondrial membrane potential (∆ψ). The gradient force draws protons through the protein ATP synthase (complex V), turning the rotor subunit (much like a water mill), and this movement is used to create ATP.

The electron transport chain (ETC) begins with the arrival of the energy-rich nicotinamide adenine dinucleotide (NADH) to respiratory complex I and flavin adenine dinucleotide (FADH_2_) to complex II. These molecules contain electrons with high transfer potential. The electrons are then shuffled from complex I to complex IV using electron carriers, releasing energy that is utilized for proton pumping and the creation of ∆ψ. The last electron acceptor is oxygen. Complex IV transfers electron to oxygen and, along with hydrogen, creates water molecules (Figure 1). Therefore, oxygen must be present for the oxidative phosphorylation process to occur, and decreased oxygen concentration (hypoxia) is a major stressor disrupting aerobic functions, especially ATP production. There are several factors influencing the oxidative phosphorylation process, such as Ca^2+^ regulation, ROS production and availability and solubility of O_2_ (which can change under different environmental conditions, development of pathology or exercise). All of these processes affect mitochondrial function and ATP production [26].

### 2.2. Oxidative Stress

Oxidative stress is an imbalance between free radicals and antioxidants in the body, which can potentially cause DNA, protein and tissue damage. In some cases, the natural processes in our cells have deleterious outcomes, such as the production of partially reduced oxygen (free radicals). Free radicals are chemically unstable and highly reactive molecules that can oxidize other molecules in the cell.

A major site for these redox reactions, potentially resulting in free radicals, is the mitochondrion, and the main source for these free radicals is diatomic oxygen which is partially reduced in the mitochondria. In the normal ETC process, electrons will be carried to complex IV on the electron carrier (cytochrome C) and with hydrogen ions and oxygen, the final electron acceptor, will produce 2 water molecules. However, aberrations in this process can occur and produce ROS. There are several reasons for this, among them electron leakage from the electron carriers. In the ETC, a fraction of the oxygen can be reduced to superoxide (O_2_•−) [27]. Most of these sites release O_2_•− into the mitochondrial matrix, and a few of them release it into the intermembrane space [28] (Figure 1). When released into the mitochondrial matrix, O_2_•− is converted to hydrogen peroxide (H_2_O_2_) by the antioxidant enzyme manganese superoxide dismutase (mnSOD). The H_2_O_2_ is then converted to water and oxygen by the antioxidant enzyme catalase, or to water by peroxidases, which also reside in the mitochondrial matrix, depending on the cell type (Figure 1). Catalase is highly expressed in heart and liver mitochondria [29,30], while in the brain, catalase is confined to peroxisomes, and its level in the mitochondria is low [31]. Brain mitochondria are enriched with the peroxidase systems GSH/glutathione peroxidase (GPx) and Trx2/peroxiredoxin 3 and 5 (Prx3 and Prx5) that can detoxify H_2_O_2_ in the presence of respiration products (i.e., NADPH). The TrxR/Trx/Prx pathway was estimated to contribute to 70–80% H_2_O_2_ removal by brain mitochondria, and the GR/GSH/GPx pathway to an additional 10–20%. The remaining 10% is handled by non-enzymatic scavenging mechanisms [32]. If O_2_•− is released to the intermembrane space, it can leave the mitochondria as ROS and cause cell damage.

ROS are a frequent byproduct of all aerobic oxygen metabolism. ROS play a detrimental role in processes such as apoptosis [33], hyperglycemia, diabetes [34], protein aggregation, neurodegeneration [35] and cancer [36] and also participate in cell signaling [37].

The conversion of oxygen to ROS is mainly a function of metabolic rate. Complexes I and III of the ETC are especially prone to electron leakage and O_2_•− production (Figure 2). Complex I activity is considered a rate-limiting step for the mitochondrial respiratory chain and is therefore an important factor in the regulation of oxidative phosphorylation. Both defective complex I activity due to lack of substrates such as pyruvate, malate, glutamate, etc. [38], and inhibition of its activity [39] elevate ROS levels. Studies on isolated complex I have shown two main ROS-producing mechanisms, involving flavin mononucleotide (FMN) at the NADH-binding site and ubiquinone (Q) at the electron carrier-binding site. FMN initiates the ETC in complex I, where it receives electrons from NADH and passes them on through a series of iron-sulfur (Fe-S) redox clusters to the first electron carrier, Q. Once FMN is fully reduced, O_2_•− can be produced; if there is a dwindling NADH pool that will not bind to this site, O_2_ will come in contact with FMN and create ROS (Figure 2) [40]. Another pathway through which complex I creates O_2_•− is formation of the free-radical semiquinone anion species (Q•−) (Figure 2). Mitochondrial Q transfers electrons between dehydrogenases and the redox pathways of the ETC. Once formed, Q•− is highly reactive and will transfer its electron to molecular oxygen to produce O_2_•−. Studies show that complex I inhibitors, such as rotenone and piericidin A, which block the transfer of electrons from the Fe-S centers to the ubiquinone pool, increase ROS production (Figure 2) [41,42,43] because NADH can react directly with O_2_ to create either H_2_O_2_ or O_2_•− (Figure 2) [44]. These findings suggest that the rate of NADH production (from the citric acid cycle) and the ratio of O_2_ to NADH are crucial. In support of this, when rotenone was added to cells overexpressing mnSOD, the rate of ROS production was significantly lower than in controls. However, when rotenone was added to cells with defective mitochondria, it did not change the rate of ROS production [45], supporting the notion that mitochondria are the main source of ROS.

An additional pivotal site for diatomic oxygen conversion to ROS is ETC complex III. At complex III Q•−, the intermediate of ubiquinol oxidation can create ROS by giving the remaining electron directly to O_2_ [41] (Figure 2). There are two sites of ROS production in complex III: the o center, oriented toward the intermembrane space and the i center, oriented toward the mitochondrial matrix [46]. At the matrix, ROS scavengers can neutralize these reactive molecules and reduce mitochondrial and cell damage. In contrast, the o site releases ROS toward the intermembrane space, away from matrix antioxidant defenses, facilitating ROS release to the cytosol.

## 3. HBOT, Mitochondrial Function and Oxidative Stress

In the last two decades, the effects of HBOT on mitochondrial function have been examined using a variety of protocols. However, these protocols apply different pressures (ranging from 1.5 ATA to 2.5 ATA), time in the chamber (1 to 4 h) and number of treatments (1 to 60), leading to different conclusions on HBOT’s effects (Table 1). Interestingly, while 1 to 5 consecutive treatments mainly lead to a reduction in mitochondrial function, 20 to 60 consecutive treatments lead to a significant improvement in mitochondrial parameters [47,48,49,50]. Animal studies have shown that following one to five treatments, HBOT induces loss of ∆ψ (indicating reduced ETC integrity) and initiation of the mitochondrial apoptotic pathway. These adverse effects do not appear when HBOT is repeated for more than 20 days, and studies that performed treatments for 20 to 30 days showed beneficial effects on mitochondrial activity and metabolism (see more details in Table 1). Hence, HBOT shows promising effects on mitochondrial activity (ETC complexes, ∆ψ, apoptosis and ATP production) and is thus predicted to improve mitochondria function in diseases and conditions that exhibit mitochondrial dysfunction, but only when the treatment is long-term.

Although HBOT is thought to lead to higher levels of oxidative stress due to excess ROS generation by the mitochondria, it is hypothesized that as part of the protective response, the cells reduce their mitochondrial activity to lower ROS production and to alleviate the oxidative stress [57]. It is therefore possible that early elevation of ROS following single (1–5) HBOTs reduces mitochondrial activity, thereby reducing additional ROS production. This might explain the previously mentioned reduction in mitochondrial activity after 1 to 5 HBOTs.

As the hyperoxic state during HBOT can increase ROS production, one can speculate that reoccurring treatments might cause excessive generation of ROS. However, until recently, it was not clear if indeed ROS accumulate following repeat HBOTs and if these ROS also have beneficial effects on the cells. As discussed below, repetitive cycles of HBOT do not necessarily cause excess generation of ROS, and it is now known that these ROS and reactive nitrogen species (RNS) also serve as signaling molecules in transduction cascades that support healing, cell survival and proliferation through the activation of major factors, such as nuclear factor erythroid 2-related factor 2 (Nrf2), HIF1α, SIRT1, vascular endothelial growth factor (VEGF) and other growth factors and hormones [58,59,60,61,62,63]. It should also be noted that in parallel to the elevation in ROS levels, antioxidant pathways are activated. Hence, the balance between free radical levels and antioxidant levels and activity will determine the extent of the oxidative stress. Bosco et al. [64] showed that under 1.5 or 2.5 ATA, changes in ROS or antioxidant scavenger activity go back to baseline within approximately a month.

Usually, for research purposes and in the clinic, HBOT is administered at between 2 to 3 ATA. Above this pressure, where the levels are referred to as supranormal, the main outcome is a clear increase in bio-oxidative products and saturation of antioxidants, leading to enhanced oxidative damage [65,66]. Exposure to high oxygen levels for a long duration can cause oxygen toxicity and may lead to systemic damages such as seizures, lung malfunction or retinopathy of prematurity [67]. The occurrence of oxygen toxicity and seizures is very low [68]; however, to minimize side effects and complications, HBOT should be provided by a certified and trained medical staff using strict operational protocols, including pre-therapy evaluations, appropriate exclusion criteria and in-chamber monitoring [68,69].

To reduce the chances of damage, exposure to HBOT in the clinic is rather brief (generally 1 h per treatment) and at pressures lower than 3 ATA, most commonly 2 ATA. In some cases, intermittent HBOT protocol is used, where oxygen levels are changed several times during a single treatment [13,70]. As a result, if HBOT is administered within therapeutic and approved limits, an antioxidation process mostly accompanies the rise in oxidation products (Table 2) and is adequate to counter the oxidative stress damage. Moreover, following the HBOT session, the reversal of oxidation products occurs approximately half an hour earlier than that of the antioxidant enzymes’ activities [71,72], which supports the safety of HBOT within therapeutic ranges.

Mechanistically, HBOT induces activation of transcription factors and gene expression, evoking antioxidant enzymatic activity, especially through the Nrf2 pathway. Nrf2 is a redox-sensitive transcription factor that is involved in cellular defense mechanisms and acts on several known target genes, including HO-1, quinone oxidoreductase 1 and glutathione S-transferase [81], all of which have detoxification properties, reducing the free radical load (Figure 2). However, activation of these transcription factors depends on the partial oxygen pressure. Mild hyperoxia (30% O_2_) initiates activation of HIF1α [82] and decreases Nrf2 levels in the nucleus [83], and administration of 100% O_2_ elevates Nrf2 levels in the nucleus and upregulation of Nrf2-regulated genes [84]. One of the major molecules supporting tolerance of the organism to oxidative damage is HO-1 [85], also called heat-shock protein (HSP) 32, a ubiquitously expressed multitasking enzyme that has neuroprotective action [85]. Under basal conditions, Nrf2 is found in a ubiquitinated state and is degraded in the proteasome (Figure 3), whereas in the presence of oxidative stress and electrophiles (reactant species willing to accept electrons), the ubiquitin E3 ligase complex is modified and Nrf2 is stabilized [86]. Once stabilized, it is transported to the nucleus where it activates its known antioxidant response (Figure 3) by acting on the aforementioned gene targets [58,87]. Indeed, after 25 HBOT sessions, tissue levels of Nrf2, along with its downstream targets, were significantly increased compared to those in nontreated control patients [88].

In summary, short-term HBOT creates stress and causes mitochondria to reduce their activity, which partially decreases ROS production. However, in long-term HBOT, antioxidant scavenger activity is elevated, helping the mitochondria function without disturbing the redox balance and even enhancing their activity.

## 4. The Interplay between SIRT1, HIF1a and ROS during HBOT

Most HBOT protocols consist of not only breathing 100% oxygen under high atmospheric pressure, but also intermittent fluctuations in oxygen [13,70]. These fluctuations occur between the daily HBOTs as oxygen levels return from 100% to 21% at the end of the daily treatment. In addition, a unique oxygen fluctuation protocol changes oxygen levels from the physiological 21% oxygen to 100% oxygen and back to physiological oxygen levels of 21% several times during each treatment [13]. These fluctuations are interpreted by the body as a hypoxic signal—lack of oxygen (moving from 100% to 21%), although hypoxia does not actually occur. These signals and fluctuations activate specific cellular mechanisms including the activation of transcription factors [89] that enhance cell activity and the body’s natural rejuvenation potential. This phenomenon, called the hyperoxic–hypoxic paradox (HHP) or the “normobaric oxygen paradox”, is key to the effect of HBOT [13,70,82,90] and is highly effective at enhancing subjects’ performance.

During such intermittent HBOT protocol, two phenomena occur sequentially and repeatedly—hyperoxia and apparent hypoxia. This recruits two additional important factors that affect mitochondrial function, mitochondrial biogenesis and oxidative stress: hypoxia-induced factor (HIF1a) and SIRT1, a class III histone deacetylase. SIRT1 belongs to a large family of sirtuins also known as longevity proteins. HIF1a is important for many processes, including the formation of red blood cells via the transcription of erythropoietin and activation of VEGF to increase blood vessel formation, both of which are needed to combat the hypoxic state [91]. The intermittent HBOT raises HIF1a activity at a systemic level [82]. In addition, elevation of oxygen activates another transcription factor, Nrf2, that regulates the expression of antioxidant proteins [70,90] as detailed above.

SIRT1 utilizes nicotinamide adenine dinucleotide (NAD^+^) as a substrate and is involved in neuronal survival, cell proliferation, metabolic dysfunction and in modulating the stress response. During normoxia, prolyl hydroxylase domain (PHD) proteins are activated and destabilize HIF1α (Figure 4). Hypoxia, through the inhibition of PHD, stabilizes HIF1α, which interacts with HIF1α to form an active HIF.

Hypoxia also reduces NADH consumption and the level of NAD+, which inhibits SIRT1 (Figure 4). Reduced SIRT1 activity reduces mitochondrial biogenesis. On the other hand, the hyperoxic state during HBOT increases NADH consumption and NAD+ levels, hence activating SIRT1 and increasing mitochondrial biogenesis. Nevertheless, the effect of the apparent hypoxia during intermittent HBOT can also inhibit SIRT1 and reduce the net effect of hyperoxia on mitochondrial biogenesis.

On a parallel but intertwined pathway, HIF1α also affects SIRT1 and is affected by ROS (Figure 4). As mentioned above, during intermittent HBOT, due to formation of excess ROS scavengers and antioxidants, the level of ROS decreases. As PHD proteins need ROS for their activity, PHD level decreases, and consequently, HIF1α level is expected to increase (even though this is not a hypoxic state). As HIF1α inhibits SIRT1, this can reduce mitochondrial biogenesis. In addition, oxidative stress elevates Nrf2 as described above [84,88]. Future studies should thoroughly investigate the interplay between various key factors, such as HIF1a and SIRT1, which are important for many processes, including neuroprotection, mitochondrial biogenesis and rejuvenation. These factors are affected not only by hypoxia, but also by the intermittent hyperoxia–hypoxia cycles that are commonly practiced in HBOT [13,70] and change according to the partial pressure subjected to the patient. A recent study suggests that at a lower pressure (30%), HIF1a is activated, but only for the adjacent time after the HBOT exposure. At a higher pressure (100%), HIF1a is also activated right after the HBOT, but above that (140%), HIF1a is no longer active [90].

Interestingly, moderate normobaric intermittent hypoxia (exposure to 10 cycle/day of 10% oxygen for 6 min for 21 days) showed improvement in the 3xTg-AD mouse model and was associated with increase in BDNF, erythropoietin (EPO) and improved spatial memory [92,93]. The exact mechanisms are still not known, but it is expected that HIF1a and the mitochondria are central in this process [94,95,96].

It should be noted that changes in SIRT1 have also been detected during and following physical exercise [97]. Similar to HBOT, physical exercise increases the consumption of oxygen and leads to elevation of ROS. This is accompanied by elevation of antioxidant and oxidative damage repair mechanisms, including the thiol antioxidants glutathione and thioredoxin. The antioxidant activity compensates for the exercise-induced ROS production and is part of the cellular and physiological processes involved in exercise [97]. Interestingly, exercise restores the level of SIRT1 in patients with neurodegenerative diseases and attenuates the severity of these diseases [98]. More research aimed to understand the effects of intermittent HBOT, physical exercise and intermittent hypoxia will lead to better understanding of the mechanisms of action and advantages and may lead to synergistic treatments or improvement in performance of these treatments.

## 5. Disease and Oxidative Stress

Oxidative stress is associated with the disease course of cancer, diabetes, neurodegenerative disorders and more. Many studies have shown that this association alters metabolic rate, which in turn contributes to development of the pathophysiology. Hyperglycemia induces oxidative stress by elevating ROS production and interfering with ROS-scavenging activity. Targets for oxidative stress in the beta cell are likely to include PDX-1, a transcription factor that plays an important role in pancreas development and differentiation, as well as in maintaining normal beta-cell function. Thus, when cells are exposed to H_2_O_2_ and elevated ROS production, PDX-1 is inhibited, and consequently, insulin gene expression is compromised [99,100]. These reactive species interact with transcription factors, increase endothelial dysfunction and dysregulate enzymatic activity, leading to vascular inflammation [101]. Diabetic patients receiving HBOT show increased level of lipid oxidation after the first treatment, yet after the 15th treatment, these patients returned to basal level (before the first treatment) [102]. The authors hypothesized that this effect could be explained by increased activation of antioxidant enzymes. In cancer cells, the redox homeostasis is altered, which renders the cells resilient to exogenous stressors. ROS can both initiate cancer and contribute to its progression [103]. Some of the anticancer drugs that are in clinical use today act by regulating oxidative stress and elevating ROS production to initiate apoptosis of the mutated cells [104]. While irradiation therapy increases oxidative stress markers (such as DNA oxidative damage), treatment of HBO reversed this effect by elevating the antioxidant SOD and HO-1 activity [76]. Other pathologies that go hand in hand with oxidative stress are neurodegenerative diseases. In Alzheimer’s disease, brain regions with high concentrations of the aggregated amyloid-β (either 1–40 or 1–42) show higher levels of oxidative stress markers [105], whereas brain regions with a low concentration of amyloid-β aggregates do not show elevation in oxidation products [106]. Yatin et al. [107] showed that knocking out methionine residue 35 on amyloid-β_1–42_ prevents the creation of free radicals in vitro, thus demonstrating the involvement of amyloid-β_1–42_ in the formation of ROS in Alzheimer-diseased brains. Other studies have shown the involvement of the methionine at residue 35 on amyloid-β in neurotoxicity [108]. Methionine is a readily oxidizable amino acid and can undergo 2-electron oxidation to form methionine sulfoxide and cause protein carbonylation [109]. In Huntington’s disease, inducing an antioxidant enzyme in the mitochondria attenuated disease pathophysiology: animals gained more weight, with decreased neuronal death and delayed motor impairments [110]. All of these pathologies (and more) present different etiologies, yet all present with alterations in ROS production. Many mutations and diseases disturb mitochondrial equilibrium which leads to further damage, aggravating the insult to the cells. Mitochondria are the major source of ROS in the cells. Nevertheless, in moderation, ROS are utilized for routine physiological functions. Researchers have been studying the effects of HBOT on neurodegenerative diseases and have found promising results. HBOT on neurodegenerative disease animal models show positive effects on the basal elevated oxidative stress levels. In these studies, HBOT was shown to alleviate the levels of ROS damage and increase antioxidant activity [53,54,111]. As of today, clinical trials focus on the advantageous effects of HBOT on cognitive decline and quality of life for patients suffering from neurodegenerative diseases and show that HBOT improve cognitive functions and ameliorate the reduced brain metabolism in MCI and AD patients [10,11,112,113,114] and in animal models [12,53,111,115,116,117]. To better understand the effects of oxidative stress and antioxidant scavenger activity in clinical trials, more research is needed.

In immune cells, mitochondrial function and metabolism were found to drive responses against bacterial infections and inflammation. ROS damages bacteria and parasites (as bacteria lack antioxidant defenses) and, by this, helps macrophages eliminate them [118]. The adaptive immune system uses ROS to regulate signal transduction by cell-surface receptors. Mitochondrial ROS in affected T cells induce activation and proliferation through secretion of Il-2 (T-cell survival marker), and the addition of antioxidants inhibits T-cell expansion [119]. Similar effects have been found in neutrophil activation. Asehnoune et al. [120] found that ROS initiate neutrophil activation. Once cells are exposed to ROS, NF-κB translocation to the nucleus is enhanced, and cytokine-induced activation of NF-κB could be prevented by antioxidants [121]. HBOT’s mechanism of action in infections is by elevation of ROS [122,123]; such elevation in oxidative stress renders the bacteria more susceptible to damage [124]. NF-κB activation also increases anti-apoptotic-related genes such as Bcl-2 in immune cells [125].

Taken together, this emphasizes the importance of the mitochondria and maintenance of mitochondrial equilibrium with the environment in avoiding oxidative stress. Imbalances can occur as a result of increased free radicals or a decrease in antioxidant defense; either way, inducing the mitochondria to restore normal levels of ROS is a vital step in treating any disease.

## 6. Summary

HBOT’s effect on oxidative stress and mitochondrial activity changes over the course of the treatment. Short-term treatment (one to five sessions) has been shown to have deleterious effects, though longer treatment (twenty to thirty sessions and more) has beneficial effects. Intermittent HBOT fluctuations induce the HHP, which increases oxidative stress scavenger transcription factors and their subsequent antioxidant enzyme production. As mitochondrial dysfunction and oxidative stress are associated with many different diseases, elevation of antioxidant activity may be a major pathway underlying HBOT’s benefit in the clinic.

## Figures and Tables

**Figure 1 biomolecules-11-01827-f001:**
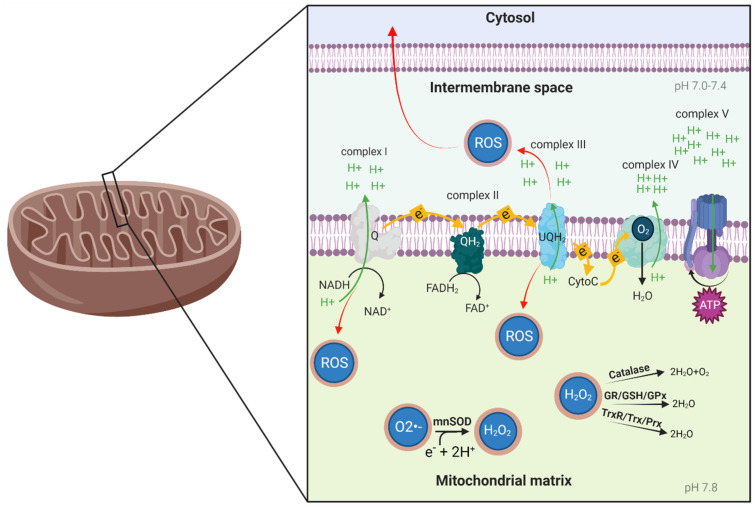
Oxidative phosphorylation process and oxidative stress balance. NADH, a product of the citric acid cycle, is oxidized to NAD^+^ at complex I, transferring an electron to ubiquinone (Q). At complex II, FADH_2_ is oxidized to FAD^+^, transferring another electron to Q. When Q arrives at complex III, it transfers its electrons to cytochrome C (CytoC). At complex IV, the electrons carried by CytoC are drawn to diatomic oxygen (O_2_) and together with hydrogen, they form H_2_O. In the electron transport chain process, complexes I, III and IV pump protons from the mitochondrial matrix to the intermembrane space, creating an electrochemical proton gradient and a membrane potential (∆ψ) on the inner membrane of the mitochondria. Complex V (ATP synthase) uses the proton gradient to allow protons back into the mitochondrial matrix, synthesizing ATP in the process. Complexes I and III are the major sites of free radical production; if they release superoxide (O_2_•−) to the mitochondrial matrix, mnSOD will transform it to H_2_O_2_. Then, depending on cell specificity, H_2_O_2_ will be transformed to O_2_ and H_2_O by catalase, or to H_2_O by glutathione reductase/glutathione peroxidase (GR/GPx) or thioredoxin reductase/peroxiredoxin (TrxR/Prx) activity, in the presence of respiration products (i.e., NADPH).

**Figure 2 biomolecules-11-01827-f002:**
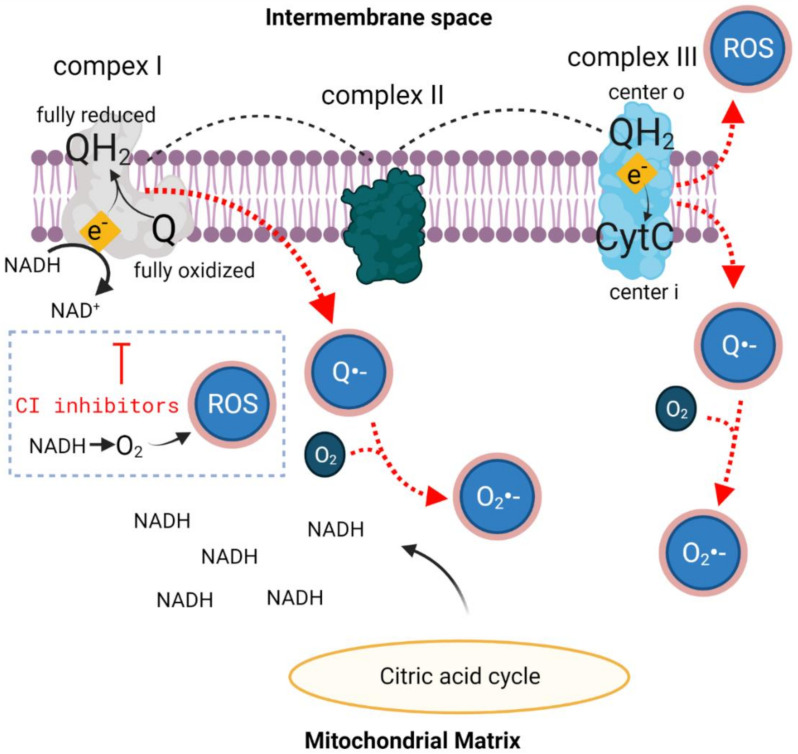
Semiquinone and ROS formation in complex I and complex III. The electron transport chain begins with the arrival of NADH to complex I from the citric acid cycle. NADH gives its electrons to ubiquinone (Q), which is the first electron carrier. This electron transfer can yield an either fully reduced ubiquinol (QH_2_)—the favorable outcome, or a semi reduced semiquinone (Q•−)—the ROS outcome. At complex III, electron transfer between QH_2_ and the next electron carrier, cytochrome C (CytC), can also result in the fully reduced CytC or Q•−. These redox reactions can yield ROS when homeostatic conditions change. Since NADH is a redox-ready molecule, it has a high potential to reduce other molecules in its surroundings. In the situation when complex I’s NADH binding site is occupied, especially when complex I inhibitors are used and there is a large pool of NADH arriving from the citric acid, NADH and O_2_ could react to create ROS. A plethora of O_2_ in the tissue, such as in HBOT, can increase interaction between O_2_ and redox-ready molecules such as NADH or Q•−.

**Figure 3 biomolecules-11-01827-f003:**
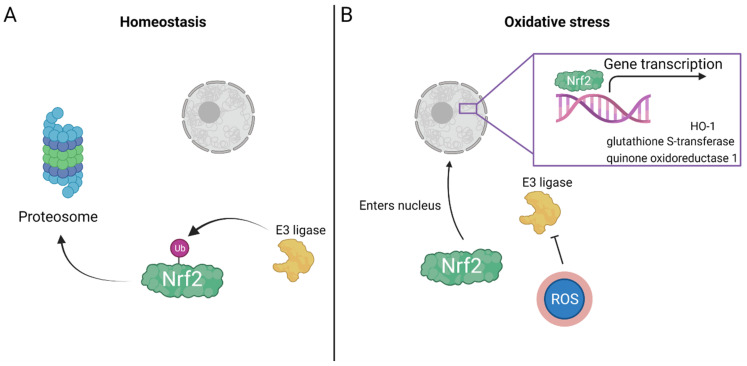
Nrf2 enters the nucleus under oxidative stress conditions. (**A**) At basal conditions, the E3 ligase targets Nrf2 for ubiquitination and marks it to be sent to the proteosome for degradation. Thus, Nrf2 does not enter the nucleus. Under oxidative stress conditions, free radicals can attach to the E3 ligase creating a conformational change that inhibits its binding to Nrf2 and, by this, allows Nrf2 to enter the nucleus. (**B**) In the nucleus, Nrf2 initiates gene transcription of protective genes, such as the antioxidants HO-1, glutathione S-transferase, quinone oxidoreductase 1.

**Figure 4 biomolecules-11-01827-f004:**
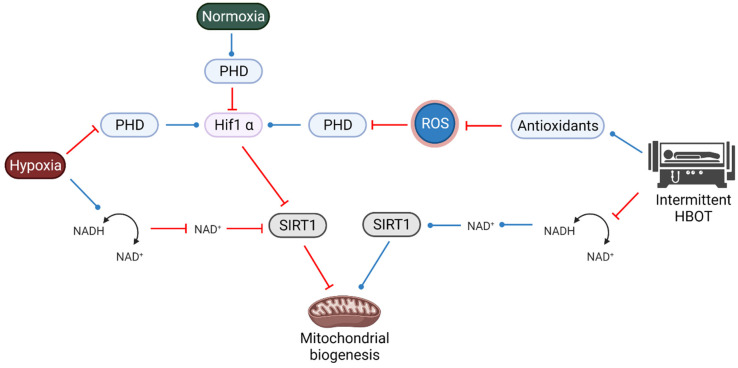
Interplay between SIRT1, HIF1a and ROS during HBOT. Under normoxia, PHD senses oxygen and leads to the degradation of HIF1a. Hypoxia inhibits PHD, leading to an increase in HIF1a, while intermittent HBOT elevates antioxidants, which leads to inhibition of PHD, therefore mimicking a hypoxic state and also activating HIF1a. In a parallel by interwind pathway, hypoxia causes inhibition of SIRT1 via changes in NADH/NAD^+^ ratio that can lead to inhibition of mitochondrial biogenesis, while intermittent HBOT activates SIRT1, leading to an activation of mitochondrial biogenesis.

**Table 1 biomolecules-11-01827-t001:** Effects of HBOT on mitochondrial activity in different pathologies and protocols.

Study	Treatment Length	Pressure	Effect on Mitochondria	Disease/Condition
Kurt et al., 2008 [51]	4 weeks	3 ATA	Increased energy production	Healthy
Dave et al., 2003 [52]	30 days	2 ATA	Improved complex IV activity	Wobbler mice (model for amyotrophic lateral sclerosis (ALS))
Tian et al., 2013 [53]	20 days of 1 h treatments	2 ATA	Reduced mitochondria-mediated apoptosis signaling (increased Bcl-2 and decreased Bax)	Amyloid-β_25-35_-injected rats
Pan et al., 2015 [54]	14 days of 1 h treatments	2.5 ATA	Reduced mitochondria-mediated apoptosis signaling (increased Bcl-2 and decreased Bax)	Rat model of Parkinson’s disease
Botigeli Baldim et al., 2013 [47]	A single treatment of 1 h	2 ATA	Reduced baseline mitochondrial consumption rate	Ischemia-induced rats
Zhou et al., 2007 [49]	A single treatment of 1 h	1.5 ATA	Increased ATP levels	Traumatic brain injury
Palzur et al., 2008 [48]	4 treatments (twice, 2 consecutive) of 45 min each	2.8 ATA	Reduction in mitochondrial membrane potential and an increase in caspase 8 activity level	Focal brain injury
Zhao et al., 2020 [55]	Twice daily 90-min treatments for either 1, 2 or 3 days	2.5 ATA	Day 1—increased mitochondrial apoptosis activation mediated by Bcl2/Bax ratio. Reduced caspase 3 and 9 activity in damaged area. Increased ATP levels. Day 2—similar results. Day 3—no activation of protein apoptosis or differences in energy production	Pancreatitis-induced rats
Han et al., 2017 [50]	Five 1 h treatments	2.4 ATA	Reduction in ∆ψ after HBOT in control group, but not in the mitophagy-inhibited group	Mitophagy-inhibited rats
Shams et al., 2017 [56]	Five 1 h treatments	2 ATA	Reduced apoptosis in HBOT rats (compared to nontreated sciatic nerve-damaged rats)	Rats with sciatic nerve damage

**Table 2 biomolecules-11-01827-t002:** Effects of HBOT on oxidative stress balance in different pathologies and protocols.

Study	Treatment Length	Pressure	Effect on Oxidative Stress	Disease/Condition
Zhou et al., 2018 [73]	One treatment of 60 min	2.7 ATA	ROS were significantly elevated in the mitochondria and in the cell	HUVEC culture
Dennog et al., 1996 [74]	A total of 75 min—three 20 min treatments with a 5 min break between	2.5 ATA	Oxidative DNA damage in the leukocytes	Healthy human volunteers
Topuz et al., 2010 [75]	One treatment of 90 min	2.4 ATA	Prevented elevation in lipid peroxidation observed in nontreated group and increased antioxidant activity	Spinal cord injury in mice
Oscarsson et al., 2017 [76]	20 treatments of 90 min	2 ATA	Elevated levels of DNA oxidation and of SOD2, HO-1 and Nrf2 expression	Irradiated rats
Matsunami et al., 2010 [77]	7 days of 2 h treatment	2.8 ATA	Elevated lipid peroxidation and decreased SOD activity in treated diabetic rats	Induced diabetic rats
Simsek et al., 2012 [65]	1, 2, 3, 4, 6 or 8 weeks of 90 min treatments	2.8 ATA	ROS and radical scavenger enzyme levels in rat brain were not significantly altered	Healthy rats
Rothfuß et al., 1998 [78]	A total of 75 min—three 20 min treatments with a 5 min break between	2.5 ATA	Increased antioxidant activity lasting for at least 1 week	Healthy human volunteers
Körpınar and Uzun, 2019 [79]	3 treatments of 1 h (within 24 h)	2 ATA	Significant elevation of lipid peroxidation (MDA) and reduced antioxidant SOD levels in the plasma	Healthy rats
	3 treatments of 1 h (within 24 h)	2.4 ATA		
	15 treatments of 1 h (within 10 days)	2 ATA	No significant change in lipid peroxidation (MDA) and antioxidant SOD levels in the plasma	
	15 treatments of 1 h (within 10 days)	2.4 ATA		
Hu et al., 2014 [80]	1–4 days of 1 h treatments twice a day	2 ATA	Decreased levels of lipid peroxidation. GPx, SOD and Gr(glutathione reductase) levels increased after 1 treatment, and these levels increased further on days 2, 3 and 4 compared to the first day	Cerebral artery occlusion

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
