# Peer review of "Hyperbaric Oxygen Treatment: Effects on Mitochondrial Function and Oxidative Stress"

_biomolecules, 2021, doi:10.3390/biom11121827_

Round 1

Reviewer 1 Report

Thank you for giving opportunity reviewing good review article. I think this article can give many information make understanding for hyperbaric physicians and scientists. I suggested some minor comments.

  1. Some references in the introduction don't seem to fit. For example, "Hence, HBOT has been used to treat many diseases and was shown to improve cognition in several brain disorders [2–10]." There are a few references that are not related to Brain disorder. Please review the reference once again.
  2. Some sentences do not have reference citation. For example, "At the cellular level, HBOT can improve mitochondrial redox, preserve mitochondrial integrity, activate transcription factors, alle-viate oxidative stress and promote neuroprotection." Please review the reference once again.
  3. When you use abbreviations, write full-term at first. For example, HBOT, DNA...
  4. The same word is used as a different expression. For example, "mitocohdrion vs. mitochondria"
  5. Are all figures in the paper newly made by authors? If some figures were cited, please cite references.

Author Response

Comment #1

Some references in the introduction don't seem to fit. For example, "Hence, HBOT has been used to treat many diseases and was shown to improve cognition in several brain disorders [2–10]." There are a few references that are not related to Brain disorder. Please review the reference once again.

Citations have been checked and corrected.

Comment #2

Some sentences do not have reference citation. For example, "At the cellular level, HBOT can improve mitochondrial redox, preserve mitochondrial integrity, activate transcription factors, alle-viate oxidative stress and promote neuroprotection." Please review the reference once again.

The sentence cited by the referee is an introductory sentence that sums up the data. Each part of it is discussed in details in part 1: “HBOT, Mitochondrial Function and Oxidative Stress” and properly referenced.

Comment #3

When you use abbreviations, write full-term at first. For example, HBOT, DNA...

All were corrected

Comment #4

The same word is used as a different expression. For example, "mitocohdrion vs. mitochondria"

In most places we use mitochondria while in a few places we used mitochondrion. Mitochondrion was used for singular while mitochondria for plural.

Comment #5

Are all figures in the paper newly made by authors? If some figures were cited, please cite references.

Yes, all figures were created with license using BioRender.com. This is acknowledged in the text.

Reviewer 2 Report

The authors are presenting a very interesting manuscript addressing an important topic which has not really been analyzed until the last years. It is of great interest and I must commend the authors for the accuracy and the appropriateness of the work.

The authors explain that HBO in a repetitive intermittent setting develops a positive hermetic reaction to ROS and RNS. The authors quote Bosco et al. 2021 together with recent data describing oxygen variation related substrates expression emphasizing the “intermittent” operational mode to elicit positive reactions. They also very adequately describe the SIRT1 expression and in parallel the mitochondrial activation and biogenesis. It is clear that the manuscript advocates intermittent hyperbaric oxygen sessions, it is may be worth to note that such SIRT1 increase is also mediated by lesser Oxygen variations (Deltas) such as physical activity : Radak Z, Suzuki K, Posa A, Petrovszky Z, Koltai E & Boldogh I. (2020). The systemic role of SIRT1 in exercise mediated adaptation. Redox Biol 35, 101467.

Radak Z, Zhao Z, Koltai E, Ohno H & Atalay M. (2013). Oxygen consumption and usage during physical exercise: the balance between oxidative stress and ROS-dependent adaptive signaling. Antioxid Redox Signal 18, 1208-1246.

In fact, as the manuscript is actually written, one may understand that “only” hyperbaric oxygen is the trigger to achieve such reactions. I’m confident that the authors do not want to mislead the reader and adding one or two simple sentences in the manuscript can re-establish equity.

Thank-you for giving me the opportunity to revise such an important manuscript.

Author Response

Comment #1

The authors explain that HBO in a repetitive intermittent setting develops a positive hermetic reaction to ROS and RNS. The authors quote Bosco et al. 2021 together with recent data describing oxygen variation related substrates expression emphasizing the “intermittent” operational mode to elicit positive reactions. They also very adequately describe the SIRT1 expression and in parallel the mitochondrial activation and biogenesis. It is clear that the manuscript advocates intermittent hyperbaric oxygen sessions, it is may be worth to note that such SIRT1 increase is also mediated by lesser Oxygen variations (Deltas) such as physical activity

We would like to thank the referee for raising this aspect. We have extended our discussion and included a new section describing similarity between HBOT and physical exercise.

Comment #2

In fact, as the manuscript is actually written, one may understand that “only” hyperbaric oxygen is the trigger to achieve such reactions. I’m confident that the authors do not want to mislead the reader and adding one or two simple sentences in the manuscript can re-establish equity.

As mentioned above, we have extended the discussion about the contribution of exercise to improvement in performances and we also added information about intermittent hypoxic treatment, to inform the readers about some alternatives.
